# Snowprint: a predictive tool for genetic biosensor discovery
Simon d'Oelsnitz [1,3] ✉, Sarah K. Stofel[1], Joshua D. Love[2] & Andrew D. Ellington [1]

Bioengineers increasingly rely on ligand-inducible transcription regulators for chemical-responsive control of gene expression, yet the number of regulators available is limited. Novel regulators can be mined from genomes, but an inadequate understanding of their DNA specificity complicates genetic design. Here we present Snowprint, a simple yet powerful bioinformatic tool for predicting regulator:operator interactions. Benchmarking results demonstrate that Snowprint predictions are significantly similar for >45% of experimentally validated regulator:operator pairs from organisms across nine phyla and for regulators that span five distinct structural families. We then use Snowprint to design promoters for 33 previously uncharacterized regulators sourced from diverse phylogenies, of which 28 are shown to influence gene expression and 24 produce a >20-fold dynamic range. A panel of the newly repurposed regulators are then screened for response to biomanufacturing-relevant compounds, yielding new sensors for a polyketide (olivetolic acid), terpene (geraniol), steroid (ursodiol), and alkaloid (tetrahydropapaverine) with induction ratios up to 10.7-fold. Snowprint represents a unique, protein-agnostic tool that greatly facilitates the discovery of ligand-inducible transcriptional regulators for bioengineering applications. A web-accessible version of Snowprint is available at https://snowprint.groov.bio.

Ligand-inducible transcriptional regulators are becoming indispensable biosensors for synthetic biology and bioengineering applications, such as high-throughput screening, dynamic regulatory circuits, and diagnostics[1–4]. Such applications typically repurpose regulators mined from diverse microbial genomes in a process called "biosensor domestication", whereby the regulator protein and the regulator-binding sequence, or operator, are retrofitted in a genetic circuit to express a heterologous gene, often in a model organism such as *Escherichia coli*[5,6].

One major challenge with this process is the identification of the regulator's cognate operator. Without knowing the precise DNA sequence a regulator binds to, bioengineers would typically resort to using a large genomic fragment hypothesized to contain the operator to drive expression of a reporter gene[6]. In consequence, this approach provides little flexibility to tune sensor performance, since cryptic promoters or inhibitory noncoding regions may produce high basal signals or dampen a sensor's responsiveness[7]. To determine minimal operator sequences for improved promoter design, widely adopted methods including electromobility shift assays (EMSA), DNase footprinting, and surface plasmon resonance (SPR) are employed[8–10]. These techniques can be used to accurately identify protein-DNA interactions with high sensitivity and resolution, but require

laborious protein purification and generally cannot be performed at scale. High-throughput alternatives have been developed that leverage next-generation sequencing, such as DNA affinity purification sequencing (DAP-seq), but these methods often require specialized infrastructure that is not easily adopted by many labs[11]. Some of these approaches do not actually delimit the relative importance of each base within the operator sequence, possibly leading to further dissection of the binding site.

By comparison, computational methods for predicting protein-DNA interactions can potentially identify binding sites. These programs typically rely on phylogenetic footprinting: aligning several homologous response elements and extracting conserved motifs that represent hypothesized operator sequences[12]. This method is embodied in tools such as Micro-footprinter, BoBro 2.0, MP[3], and DMINDA 2.0, but these implementations are oftentimes limited to a constrained set of genomes, have low predictive accuracy, and may require extensive input from the user to create a prediction[13–16]. Alternatively, statistical learning and structural information has also been used to create models capable of predicting transcription factor binding sites, but these models cannot generalize across diverse protein families[17]. As over 71 transcription factor structural families have been reported to date[18], a generalizable tool for predicting binding sequences

[1]Department of Molecular Biosciences, University of Texas at Austin, Austin, TX 78712, USA. [2]Independent Web Developer, Bentonville, AR 72712, USA. [3]Present address: Synthetic Biology HIVE, Department of Systems Biology, Harvard Medical School, Boston, MA 02115, USA. ✉e-mail: simonsnitz@gmail.com

would enable biotechnologists to identify and repurpose many more transcription factors.

To facilitate the domestication of biosensors for engineering applications, we developed Snowprint, a protein-agnostic prediction tool that identifies inverted repeat-containing operator sequences for transcriptional regulators. Benchmarking results demonstrate that Snowprint predictions are significantly similar for 67 out of 147 experimentally validated regulator:operator pairs from diverse organisms that span five distinct regulator structural families. To demonstrate practical utility for engineering, Snowprint was used to predict the operators of 33 uncharacterized TetR regulators across 10 unique phylogenetic classes, of which 28 were able to effectively repress GFP expression within *E. coli* and 24 achieved over a 20-fold dynamic range. The top 24 newly identified regulatory pairs were screened for response to industrially relevant biosynthetic intermediates, and novel genetic biosensors for olivetolic acid, geraniol, ursodiol, and tetrahydropapaverine were found, achieving induction ratios of 6-, 3.6-, 2.3-, and 10.7-fold, respectively (see Supplementary Note 1 for definitions). A simple input (one protein accession ID) and interactive data visualization interface should facilitate the adoption of Snowprint among synthetic biologists and biotechnologists.

## Results
### Workflow for operator prediction
To create a computational tool that would be capable of operator prediction, we designed a bioinformatic algorithm based on logical principles: first, that transcriptional regulators are often autoregulatory and therefore bind to DNA sequences within their own promoter[19]. Second, that transcriptional

regulators often exist as dimers and bind to inverted repeat sequences[20]. Third, that operator sequences are much more conserved than non-regulatory regions[21].

To enable operator prediction, we created a workflow that starts with a regulator accession ID from the NCBI protein database, which is in turn used to fetch (1) the regulator's protein sequence and (2) the local genetic context of the regulator, including upstream and downstream genes that are predicted to belong to the same operon as the regulator (Fig. 1). We nominally assume that adjacent genes that share the same orientation are co-transcribed with the regulator itself and in consequence, their shared promoter region often contains the regulator's cognate operator[19]. A sequence located between two divergently expressed sets of genes (the inter-operon region) is extracted, and the algorithm searches for inverted repeat sequences, using a scoring function that favors longer repeats, yet allows for imperfect and discontinuous repeats often found in nature[22] (see Methods). The sequence with the highest inverted repeat score, which we term a "seed operator", is then used for the subsequent step.

Since true operators will likely be highly conserved, the seed operator is compared to inter-operon regions of regulator protein homologs collected using BLAST, extracted in a similar manner. All inter-operon regions are aligned, sequences homologous to the seed operator are extracted, and these sequences are used to create a consensus predicted operator that contains the most frequently occurring nucleotide over the homologous sequences for each position in the operator. A corresponding conservation score is created by averaging the relative levels of enrichment for each nucleotide in the predicted operator, indicating the degree of conservation along the whole predicted operator sequence (see Methods). The consensus operator

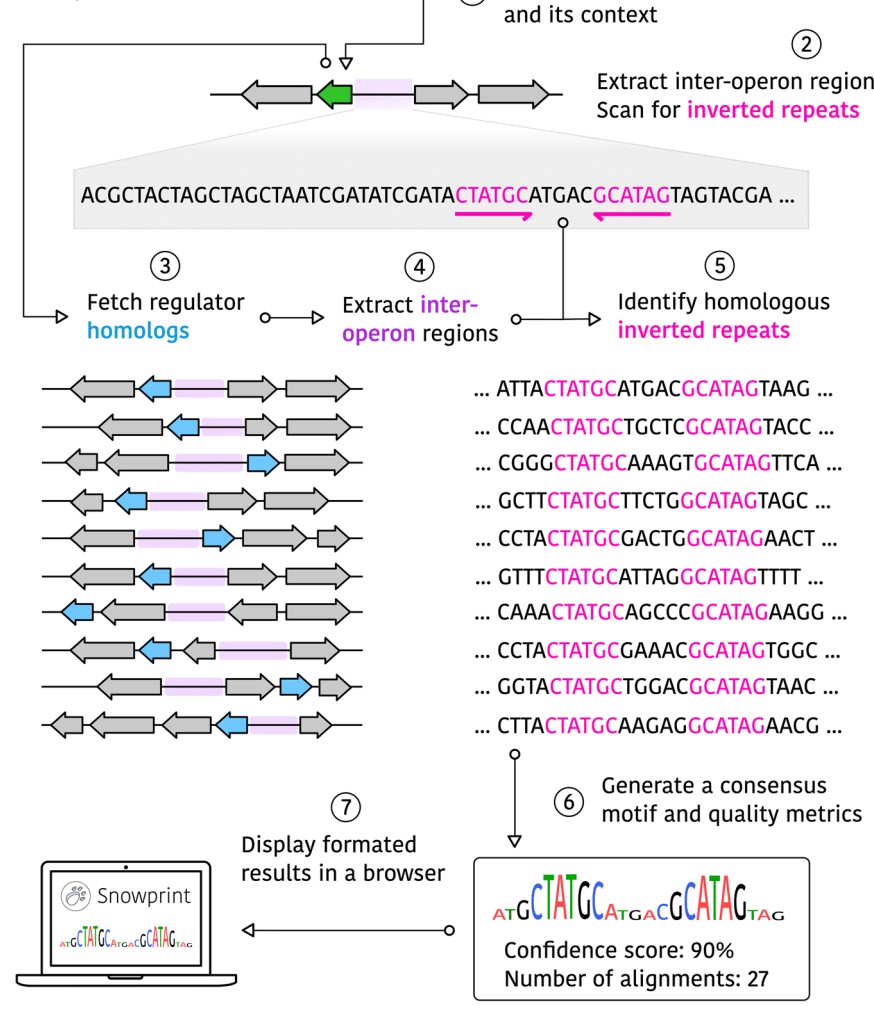

**Fig. 1 | The Snowprint workflow.** A RefSeq or GenBank accession ID is used to fetch the protein sequence of the regulator and the DNA sequence of the local genetic context (1). The inter-operon region, predicted to contain the regulator's corresponding operator, is then scanned for inverted repeat sequences (2). BLAST is then used to collect regulator homologs (3), which are used to collect homologous inter-operon regions (4) that are also scanned for inverted repeats similar to that found for the original regulator (5). The homologous inverted repeat sequences are then used to create a consensus sequence, representing the predicted operator, and associated metrics (6), which are displayed in a browser (7). The Snowprint logo was designed using a vector graphics editor.

and conservation scores are displayed in a browser using an interactive React webpage (Supplementary Fig. 1), or via a web-accessible version available at https://snowprint.groov.bio. Altogether, this algorithm design is encoded in a program we call Snowprint, a lightweight computational method to determine a regulator's DNA "footprint".

## Benchmarking Snowprint performance

To benchmark the performance of Snowprint, we first curated a list of 147 experimentally validated regulator:operator pairs from the literature (Supplementary Data 1). Operators for regulators belonging to the TetR, LacI, MarR, IclR, and GntR structural families were chosen based on the identification of an inverted repeat (Fig. 2a). A separate group ("Other") containing regulators from the ArsR, MerR, TrpR, and ROK structural families was also included for comparison. Key metrics used for benchmarking analysis included E-values generated from aligning predicted operators to known operators, as well as the conservation score, which indicates the level of sequence conservation within the predicted operator. Operator predictions were considered to be significant if the E-value produced by aligning the predicted sequence to the known operator was less than 0.01 (see Supplementary Table 1). The inverted repeat score (see "Methods") and the number of homologs used to generate the predicted operator were also collected and indicate a quality level for each prediction, since the confidence in a prediction is correlated with the number of homologs assessed and the length of the inverted repeat (Supplementary Data 1).

We found that Snowprint was able to identify operators that were significantly similar to known, experimentally validated operators

($E$-value < 0.01) for 58%, 50%, 44%, 42%, and 35% of regulators across the TetR, IclR, MarR, LacI, and GntR structural families, respectively (Fig. 2b). Although many existing tools for predicting transcription factor binding sites could not be compared to Snowprint due to substantial differences in the format of the input data (see Supplementary Note 2), we were able to compare Snowprint to a recently described statistical model train to predict TetR binding sites[17]. Our analysis indicates that the predictive accuracy of Snowprint was similar to the statistical model, with 31/50 operators producing an $E$-value of less than 0.01, compared to 29/50 for Snowprint (Supplementary Fig. 2, Supplementary Data 2). However, in some cases Snowprint outperformed the statistical model, even though the latter was specifically trained on TetR-family members (Supplementary Fig. 2a). Furthermore, the statistical model failed to return a prediction for proteins that did not belong to the TetR family, which was not the case for Snowprint (Fig. 2b).

Importantly, Snowprint was able to generalize across regulators from divergent phylogenetic spaces. The benchmarking dataset comprised regulators from nine distinct phylum, with the majority belonging to the *Bacillota*, *Actinomycetota*, and *Pseudomonadota* phyla (Fig. 2c). The motifs generated by Snowprint often showed high conservation in the middle of half site repeats and tapered off towards the middle and edges of the motif (Fig. 2d). The inverted repeat score of predicted operators was higher than that of the corresponding native operator (Supplementary Fig. 3). Since operators with higher symmetries have been shown to increase regulator affinity, in some cases Snowprint may prove useful as a tool to reduce background signal in genetic circuits[23,24]. Similarly, Snowprint was also able

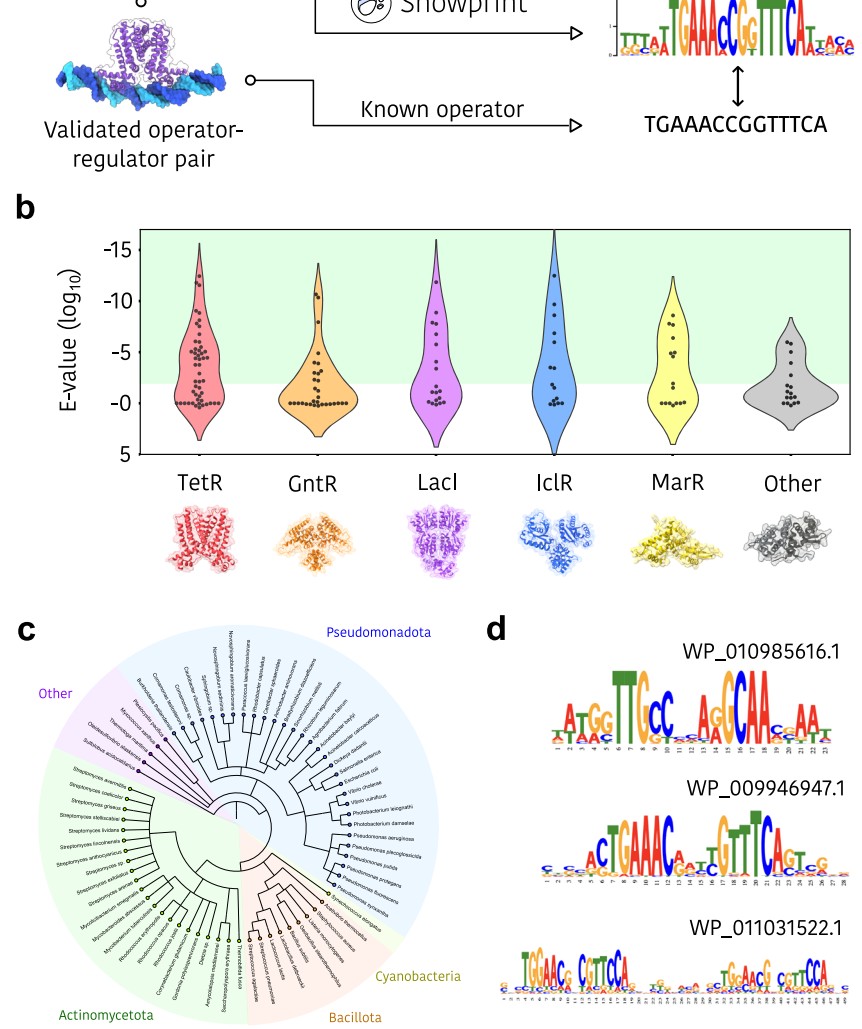

**Fig. 2 | Benchmarking Snowprint. a** Benchmarking workflow. Experimentally validated operator regulator pairs are collected from the literature, and regulators for each pair are passed through the Snowprint workflow. Predicted operators are then compared to validated operators. The regulator is colored purple and the operator is colored blue (**b**) Similarity scores for predicted operators among several structural regulator families. The *E*-value of 0.01 was used as a threshold to indicate significance (Supplementary Table 1). The "Other" group contains regulators from the MerR, ArsR, PadR, TrpR, and ROK structural families. **c** Phylogenetic diversity of the benchmarking dataset. Phylogeny.fr[52] and iTOL[53] were used to generate the phylogenetic tree graphic. Separate phyla are color coded and labeled. **d** Representative examples of predicted operator motifs generated by Snowprint. The Snowprint logo was designed using a vector graphics editor.

to identify dual inverted repeats for several regulators, which may reveal optimal spacing between operator pairs (Fig. 2d, bottom). These conservation patterns may be useful in guiding genetic designs by flagging conserved regions as indispensable and non-conserved regions as dispensable[7,23,25].

## Using Snowprint to domesticate generalist transcription factors

To demonstrate utility for synthetic biology applications, we used Snowprint to extract and "domesticate" a panel of transcription factors. We targeted TetR-family transcription factors that regulate multidrug efflux pumps in particular, since they are likely to promiscuously bind to a wide range of structurally diverse ligands and may serve as excellent starting points for directed evolution of effector specificity[26–28]. To generate designs,

TetR-family regulators were downloaded from the UniRef50 database, clustered into 30% identity groups using CD-HIT, and filtered for regulators with sequence lengths between 140-260 amino acids, typical for the TetR-family[29] (Supplementary Fig. 4). Regulators were further filtered on the basis that their annotation contained the word "regulator" and that they were adjacent to genes annotated as multidrug efflux pumps. Snowprint was then used to predict operators for the resulting regulators.

Thirty three predicted operators with the highest conservation scores were used for experimental validation (Fig. 3b, Supplementary Table 2). To test the ability of the predicted operator:regulator pairs to control gene expression in vivo, reporter plasmids were designed that put GFP under the control of the predicted operator, which was in turn placed immediately downstream from the −10 box of the *E. coli* sigma 70 promoter, known to

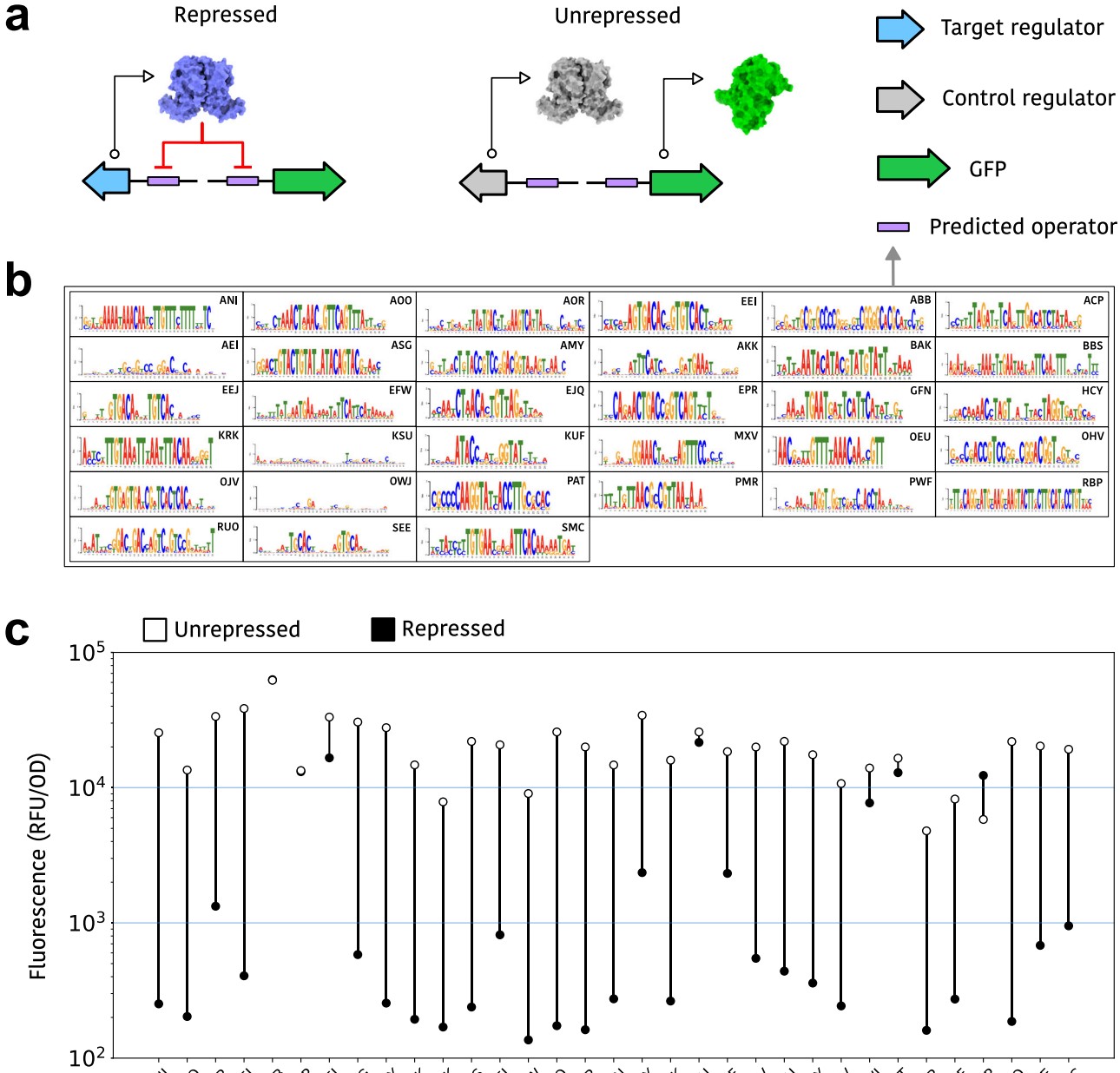

**Fig. 3 | Domestication of mined TetR regulators using Snowprint. a** Schematic of genetic circuits used to assess operator predictions. The "Repressed" circuit expresses the mined regulator and GFP, both under the control of a promoter containing the operator predicted for that regulator. The "Unrepressed" circuit differs from the former in that it expresses a control regulator, CamR, in place of the mined regulator. **b** Operator motifs generated by Snowprint for all mined regulators. Sequence motif logos were generated using LogoJS[50] (**c**) Fluorescence of *E. coli* cells expressing either the repressed or unrepressed circuits for each regulator. Data represents the mean of three biological replicates. Equivalent data displaying individual data points and standard deviation is found in Supplementary Fig. 5.

allow for tight regulation and flexible positioning to accommodate operators of various lengths[30] (Supplementary Fig. 5). The codon-optimized gene for a given transcription factor was then expressed on a separate plasmid, again under the control of the predicted operator, creating an autoregulatory circuit that should avoid potential toxic effects of regulator overexpression. To measure unrepressed synthetic promoter activity, while controlling for the cellular response to heterologous regulator expression, a control plasmid was introduced that expressed the CamR regulator (Fig. 3a); since CamR should not bind to the predicted operator, it should not repress GFP production.

Upon testing all 33 predicted operators with their cognate regulators, 28 were able to reduce the level of expressed GFP by at least 50%, and 24 were able to reduce signal by over 20-fold (Fig. 3c, Supplementary Fig. 6). This ~85% success rate is quite surprising, given that the regulators were sourced from wildly divergent microbial hosts and that synthetic promoters were crafted in accord with simple rules. Furthermore, in vivo repression could fail for reasons other than correct regulator:operator matching, such as issues relating to heterologous protein production.

### Discovering novel biosensors for biomanufacturing-relevant ligands

While Snowprint provides new potential regulator:operator pairs, the identification of appropriate ligands to control gene expression remains a daunting task. That said, the high ligand promiscuity expected by some of the newly discovered regulators suggested that an untargeted approach might actually allow the discovery of biosensors responsive to useful ligands chosen on the basis of their biomanufacturing relevance, rather than on any putative or predicted natural relevance. The compounds olivetolic acid, geraniol, sitagliptin, ursodiol, tetrahydropapaverine, and artemisinic acid at 100 uM concentration were used as test ligands, as each one is a pharmaceutical, or intermediate thereof, that has been produced from an engineered microbe or enzyme[26,31–35]. In addition, we chose these compounds due to their lipophilicity, since highly promiscuous regulators tend to have large hydrophobic binding cavities[36]. To our knowledge, natural biosensors have not been identified for any of these ligands, and in the future it is unlikely that it will immediately prove possible to regularly identify natural regulators for any biomanufacturing-relevant compound of interest.

Twenty four of the regulator:operator pairs with the highest dynamic ranges (>20-fold, average of 64-fold) were screened for de-repression (see Supplementary Table 2). Significant responses were observed for four of the six target ligands that occupy diverse structural groups, include the terpene (geraniol), polyketide (olivetolic acid), steroid (ursodiol), and alkaloid (tetrahydropapaverine) classes, with seven regulators displaying some response for at least one ligand (Fig. 4a, Supplementary Fig. 7). While the initial responses were in the 1.25- to 3-fold range, the fact that any of these uncharacterized regulators displayed a response was surprising. These results highlight that an increasing stable of new regulator:operator pairs supports the identification of biosensors for a diversity of chemical effectors.

To further validate these newly identified interactions, dose-response measurements were carried out with those pairs that produced the highest signals with geraniol, olivetolic acid, ursodiol, and tetrahydropapaverine. Tested ligand concentrations ranged from 10 uM to 5 mM, depending on the ligand's solubility limit in 1% DMSO. Sigmoidal transfer functions characteristic of transcriptional regulators were observed (Fig. 4b–e), and induction ratios reached 10.7-, 6.0-, 3.6-, and 2.3-fold for tetrahydropapaverine, olivetolic acid, geraniol, and ursodiol, respectively.

### Discussion

Snowprint is a protein-agnostic tool that leverages rapidly growing public databases of genome and protein sequences to predict transcriptional regulator:operator interactions. In contrast to existing computational tools for predicting operators, Snowprint requires minimal user input (just a single protein identifier) and is generalizable across proteins from diverse organisms and from distinct structural families[13,14]. Snowprint was able to accurately predict operator sequences for 67 of 147 experimentally-validated

regulator:operator pairs across the TetR, LacI, IclR, MarR, and GntR structural families. Taken together, these data suggest that Snowprint may be able to completely bypass traditional laborious methods for determining a regulator's operator sequence, such as EMSA and DNase footprinting, rapidly accelerating the pace of regulator discovery.

Indeed, the benchmarking results may provide a conservative estimate of predictive accuracy, since many transcriptional regulators have been shown to semi-specifically bind numerous DNA sequences and may control the expression of several genes[20,37]. Subsequent experimental validation of Snowprint predictions indicated that the predictive accuracy of the model was even higher than benchmarking results indicated. Among the 33 predictions used to create GFP reporter circuits in *E. coli*, 28 were able to modulate gene expression by over 1.5-fold, among which the top 24 circuits produced a dynamic range over 20-fold. These results were especially surprising, given that the prediction quality metrics, such as number of homologs used and conservation scores, were generally worse for the domesticated sensors relative to the validated regulator:operator pairs in the benchmarking dataset (Supplementary Table 2, Supplementary Data 1). Among the five regulators that showed no ability to repress transcription, it is possible that the Snowprint-predicted operator was not bound by the regulator, or alternatively, the regulator might not have expressed or folded appropriately in *E. coli*. Interestingly, one regulator (RBP44292.1) produced an inverted response, similar to activator proteins. This may be due to making direct contacts with RNA polymerase or via altering DNA topology to promote transcription, which has been observed in related transcription factor families[38,39].

One factor that may limit predictive accuracy is the assumption that the target regulator binds to its own promoter, which is true for the majority of prokaryotic regulators, but not all[37]. Nonetheless, this assumption is generally valid and useful in the context of identifying new regulators and operators: we compared how well both known and predicted operators mapped to upstream, inter-operon regions. This analysis revealed a coarse correlation (Supplementary Fig. 8) suggesting that if the known operator is found in the upstream, inter-operon region, then the chances of it being correctly predicted increase. Upon further analysis, the most frequent failure mode was caused by the operator sequence not being found within the extracted inter-operon region (48/80), followed by the search algorithm not identifying the appropriate operator sequence (22/80). The least frequent failure mode was caused by low quality predictions resulting from too few homologs being available (10/80) (Supplementary Fig. 9). While not encountered during benchmarking, another possible failure mode is that the regulator's genetic context might not be available. In this case a highly homologous regulator may be used as an input, which will likely produce a nearly identical operator prediction as the target regulator, since the collection of homologs used to generate the predicted operator motif will be much the same. To further enhance performance the search algorithm can potentially be improved by adding the option to search for direct repeats rather than inverted repeats[40]. Moreover, as the number of validated operators expands, "seed sequence motifs" can be used as queries, potentially avoiding the identification of false positives in the form of terminators or binding sequences of other DNA-binding proteins. As computational complexity increases, search times can be kept swift (<1 min) by fetching from locally stored databases and using more performant sequence alignment tools, such as DIAMOND[41].

Successes with Snowprint reveal that it may now be possible to regularly identify regulators for ligands that otherwise have no natural counterparts. Untargeted screening of our newly domesticated panel of generalist regulators identified binding partners for the biomanufacturing relevant compounds olivetolic acid, geraniol, ursodiol, and tetrahydropapaverine. Since the newly discovered biosensors are proximal to multidrug efflux pumps, rather than molecule-specific metabolic pathways, traditional guilt-by-association algorithms would not have been able to predict these biosensor-ligand interactions[42,43]. This untargeted discovery model complements a similar approach to biocatalyst identification, whereby panels of substrate-promiscuous enzymes are screened for a desired activity[44].

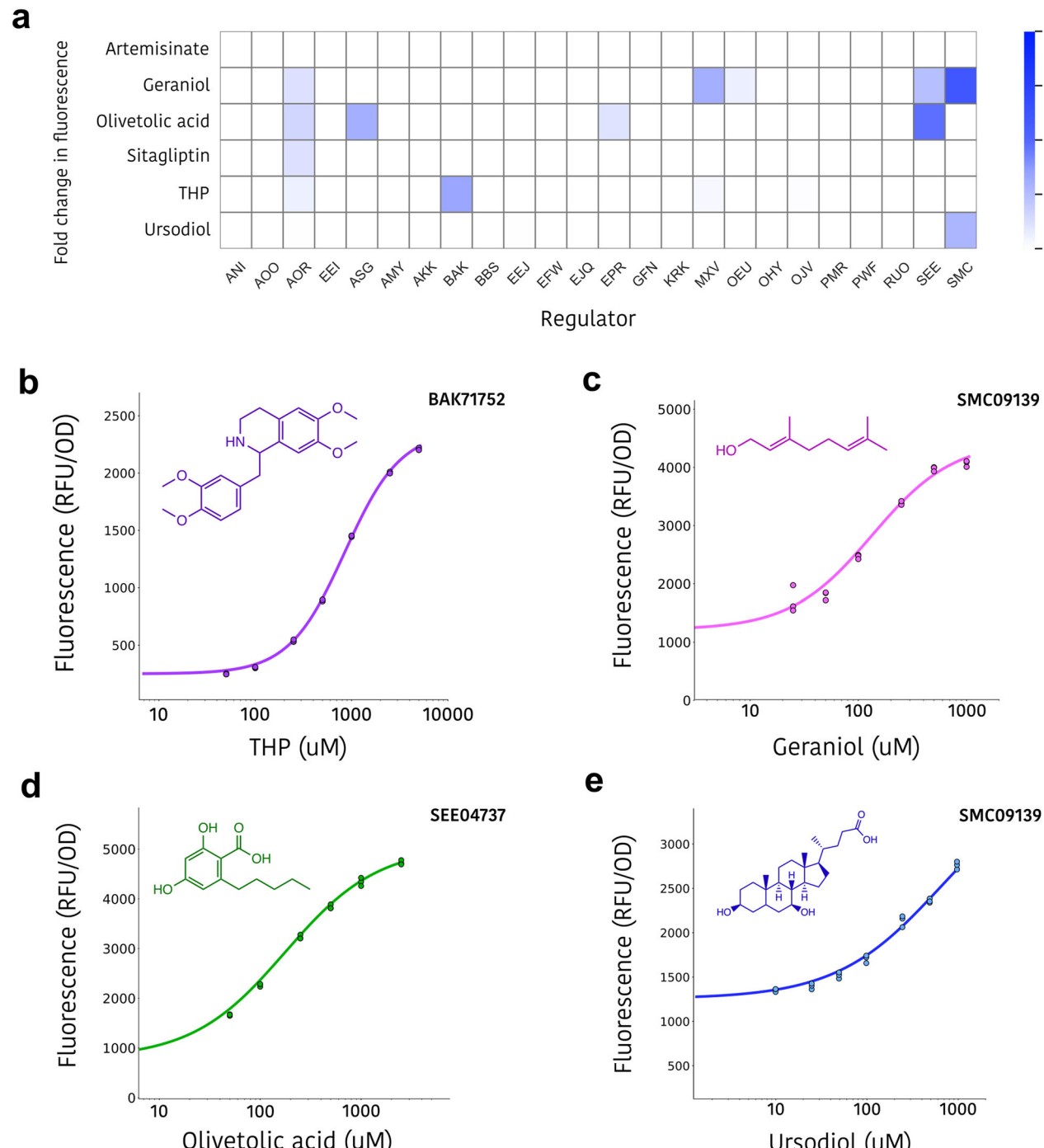

**Fig. 4 | Discovery of novel genetic sensors for biomanufacturing-relevant ligands.**
**a** Twenty four newly domesticated regulators were separately induced with 100 uM of six different ligands dissolved in DMSO in *E. coli*. Data represents the induction ratio in fluorescence over *E. coli* cells bearing the identical plasmids induced with DMSO only, performed in biological triplicate. Equivalent data displaying individual data points and standard deviation can be found in Supplementary Fig. 7.

**b–e** Dose response measurements for the BAK71752.1, SMC09139, SEE04737, and SMC09139 regulators with tetrahydropapaverine (THP), geraniol, olivetolic acid, and ursodiol, respectively. The ligand concentration was chosen based on the compound's solubility limit in 1% DMSO. Assays were performed in biological triplicate and individual data points are shown.

In the future, pairing this discovery workflow with directed evolution to refine the specificities of biosensors identified by screening[26–28] provides a new paradigm research and development teams can adopt to leverage biosensor-enabled screens for a wider diversity of small molecules. In particular, the biosensors already identified herein can be used as relevant evolutionary starting points for important pharmaceutical biomanufacturing efforts, as olivetolic acid is a precursor to all cannabinoids[33], geraniol is a precursor to all monoterpene indole alkaloids[45], tetrahydropapaverine is a precursor to three licensed non-depolarizing muscle relaxants[26], and ursodiol is directly used in the clinic to dissolve gallstones and treat liver diseases[35].

Overall, Snowprint represents a unique and, more importantly, generalizable tool that greatly facilitates the discovery of ligand-inducible transcriptional regulators. We anticipate that this tool – along with others[42,46,47] -- will be of great utility for synthetic biologists and bioengineers working to create, improve, or adapt genetic biosensors for applications in

high-throughput screening, diagnostics, and genetic circuitry[48]. Snowprint and derivatives thereof may ultimately prove capable of creating large regulator:operator datasets that can be used to train machine learning models as part of expansive efforts for more accurately predicting and generating functional protein:DNA interactions de novo[49].

## Methods

### Scoring calculations
The consensus score for an operator prediction was made using the following formula. $c = \frac{\sum_{o=0}^{x} b^2}{x}$ where $c$ = the consensus score, o = the position along the length of the operator, $b$ = the relative frequency of the dominant base within the predicted operator, and $x$ = the length of the predicted operator. The inverted repeat score for an operator sequence was made using the following formula. $s = 2x + 2y + p$, where $s$ = the inverted repeat score, $x$ = the number of matches within an inverted repeat, $y$ = the number of mismatches within an inverted repeat, and $p$ = the gap adjustment score, which adds or removes from the score based on the distance between repeat half sites as follows (0–4 bases = +4; 5–6 = +2; 7–8 = 0; 9–10 = −2; 11–12 = −4; 13–14 = −6; 15–16 = −8; 17–18 = −10).

### Benchmarking
Operator-regulator pairs were collected from the literature for structurally diverse ligand-inducible transcriptional repressors on the basis that either EMSA, DNase footprinting, or SPR were used to determine binding sites, as these are typically the most common and reliable methods for identifying transcription factor binding sites. From this dataset, the RefSeq or GenBank ID for all regulators were used as inputs to the Snowprint program, and predictions as well as prediction metrics were collected (Supplementary Data 1). The NcbiblastnCommandline function of Biopython was then used to compare the sequence similarity between the experimentally validated operator and the Snowprint-predicted operator for each regulator in the dataset, generating $E$-values using the Smith-Waterman algorithm with standard parameters (gap open penalty: 10; gap extend penalty: 0.5). An $E$-value below 0.01 was used to indicate a significant match between validated and predicted operators, since predicted operators with an $E$-value below 0.01 were found to match well with the corresponding documented operators (see Supplementary Table 1). The pairwise2 algorithm of Biopython was used to generate alignment scores for all other sequence comparisons.

### TetR dataset curation
As outlined in Supplementary Fig. 3, the UniRef50 database was used to fetch TetR-family regulators using the search query "tetr regulator". CD-HIT was subsequently used to cluster the resulting sequences into 30% identity groups. The Galaxy software suite was then used to filter out regulators with sequences above 260 amino acids and below 140 amino acids. Next, a python script was used to filter out sequences that did not contain the word "regulator" in their annotation name, and the Entrez API was then used to extract only the regulators that were located adjacent to proteins annotated as multidrug efflux pumps. Snowprint predictions were generated for the remaining regulators, and the top 33 with the best consensus scores were used for in vivo experimental validation.

### Strains, plasmids and media
*E. coli* DH10B (New England Biolabs) was used for all routine cloning and directed evolution. All biosensor systems were characterized in *E. coli* DH10B. LB Miller (LB) medium (BD) was used for routine cloning, fluorescence assays, and orthogonality assays unless specifically noted. LB with 1.5% agar (BD) plates were used for routine cloning. The plasmids described in this work were constructed using Golden gate assembly and standard molecular biology techniques. Schematics of the Regulator and Reporter vector designs used in this study are displayed in Supplementary Fig. 10. Chemical transformation was performed as follows. Briefly, 5 mL of an overnight culture of DH10B cells was mixed with 500 ml LB and grown at 37 °C and 250 r.p.m. for 3 h. Resulting cultures were centrifuged (3500 g, 4 °C, 10 min), and pellets were washed with 70 mL of chemical competence buffer (10% glycerol, 100 mM $CaCl_2$) and centrifuged again (3500 g, 4 °C, 10 min). Pellets were then resuspended in 20 mL of chemical competence buffer. After 30 min on ice, cells were divided into 250 μL aliquots, flash frozen in liquid nitrogen, and stored at −80 °C until use. This method has been described previously[26]. Synthetic genes, obtained as eBlocks, and primers were purchased from IDT. Full sequences of reporter and regulator plasmid are provided in Supplementary Figs. 11, 12.

### Chemicals
Tetrahydropapaverine was purchased from Tokyo Chemical Industry Co. (N0918). Ursodiol was purchased from MP Bio (0215825201). Geraniol was purchased from Tokyo Chemical Industry (G0027). Sitagliptin was purchased from Ambeed (654671-77-9). Artemisinic acid was purchased from Cayman Chemical (25059). Olivetolic acid was purchased from Cayman Chemical (26282).

### Biosensor response assays
Regulator and reporter plasmids were co-transformed into *E. coli* DH10B cells, which were subsequently plated on LB agar plates containing appropriate antibiotics. Three separate colonies were picked for each transformation and were grown overnight. The following day, 20 μL of each overnight culture was then used to inoculate separate wells in a 2-mL 96-deep-well plate (Corning, P-DW-20-C-S) sealed with an AeraSeal film (Excel Scientific) containing 900 μL LB medium. After 2 h of growth at 37 °C, cultures were induced with 90 μL LB medium containing either 10 μL DMSO or the target inducer molecule dissolved in 10 μL DMSO. For dose response measurements, different concentrations of the target molecule were prepared in the same format (10 uL of DMSO in 90 uL LB). Cultures were grown for an additional 4 h at 37 °C and 250 r.p.m. and subsequently centrifuged (3500 g, 4 °C, 10 min). Culture supernatant was removed, and cell pellets were resuspended in 1 mL PBS (137 mM NaCl, 2.7 mM KCl, 10 mM $Na_2HPO_4$, 1.8 mM $KH_2PO_4$, pH 7.4). One hundred microliters of the cell resuspension for each condition was transferred to a 96-well microtiter plate (Corning, 3904), from which the fluorescence (excitation, 485 nm; emission, 509 nm) and absorbance (600 nm) were measured using the Tecan Infinite M1000 plate reader. This assay format was used to determine regulator dynamic range (Fig. 3), as well as regulator screening and dose-response measurements (Fig. 4).

### Statistics and reproducibility
All data in the text are displayed as mean ± standard deviation unless specifically indicated. All experimental assays were performed in biological triplicate, which represent three individual bacterial colonies picked from an agar plate. Bar graphs, endpoint fluorescence measurements, and dose–response functions were all plotted in Python 3.10.6 using Matplotlib and Seaborn. Dose–response curves and $EC_{50}$ values were estimated by fitting to the Hill equation $y = d + (a − d)xb(cb + xb)−1$ (where y = output signal, b = Hill coefficient, x = ligand concentration, d = background signal, a = maximum signal and c = $EC_{50}$), with the scipy.optimize.curve_fit library in Python.

### Building the Snowprint web application
The web application was split into frontend design and backend data architecture. The frontend was written in Javascript using the React and Material UI libraries. Sequence logos are generated using LogoJS[50]. The backend ported all code from the command line tool used for benchmarking (https://github.com/simonsnitz/Snowprint) into a docker container hosted on AWS Fargate. The only difference between the command line tool and the web application is that the former uses NCBI's BLAST to collect protein homologs while the latter uses DIAMOND, which performs faster. While results returned from the command line tool and web application are similar, they may not be identical.

## Reporting summary

Further information on research design is available in the Nature Portfolio Reporting Summary linked to this article.

## Data availability

The relevant data are available from the corresponding author upon request. The source data for Figs. 2b, 3c, 4a–e are provided in Supplementary Data 3. NCBI RefSeq identifiers for all mined regulators in this study can be found in Supplementary Table 2. The GenBank ID for the CamR protein used as a control regulator is BAA03510.1.

## Code availability

The source code and detailed instructions for use of Snowprint are maintained in the GitHub repository located at https://github.com/simonsnitz/Snowprint. The Snowprint repository was deposited in[51]. Snowprint is Open Access under an MIT License. Source code for the frontend and backend of the Snowprint web application (snowprint.groov.bio) are maintained in the GitHub repositories located at https://github.com/simonsnitz/snowprint-ui and https://github.com/simonsnitz/snowprint-backend, respectively. Code used to generate bar plots, heatmaps, and dose–response functions presented in this manuscript is accessible in the GitHub repository located at https://github.com/simonsnitz/plotting.

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

## Acknowledgements

Funding from National Institute of Standards and Technology (70NANB21H100), the Air Force Office of Scientific Research (FA9550-14-1-0089), the Welch Foundation (F-1654), and the National Institutes of Health (R01EB026533) is acknowledged.

## Author contributions

The Snowprint command line tool was developed and benchmarked by S.D. The Snowprint web application was developed by J.D.L. S.D. designed all experiments. S.D. and S.K.S. performed biosensor measurements. The manuscript was written by S.D. with support from A.D.E.

## Competing interests

The authors declare no competing interests.
