## [Peer Review File · Communications Biology]

Reviewers' comments:

Reviewer #1 (Remarks to the Author):

The article by d'Oelsnitz et al reports a computational method for identifying potential bacterial FT-target promoter pairs that can be leveraged for designing biosensors for small (and not so small) molecules of interest. The background and methodological basis aligns well with similar approaches published in the last few years (see eg <https://doi.org/10.1021/acssynbio.2c00382>, <https://doi.org/10.1021/acssynbio.2c00491>, <https://doi.org/10.1021/acssynbio.2c00679>, <https://doi.org/10.3389/fbioe.2023.1118702>) and therefore the article is one more addition to the arsenal of choices that metabolic engineers and genetic circuit designers have for assembling given constructs.

Overall, the article is well written and the message clear (with some flaws, see below). The platform is experimentally validated and, at least in the examples presented, seems to work to an extent. There are some issues, however, that need to be tackled:

1. In the general description of approach (summarized in Fig. 1), it is taken for granted that proximity of a TF gene to a metabolic cluster is an indication that such TF regulates the gene or operon in response to pathway substrates. While this is true in many cases, there are others where this does not happen (see eg a classical discussion on the matter in <https://shorturl.at/bKXY2>). This may explain the somewhat poor predictions (67 out of 147) that could be validated upfront.
2. The platform predicts nothing re dynamic range of the transfer input-output functions, an essential parameter for circuit design. In fact the data of Fig. 4 shows very different ranges, which may or not be useful for the envisioned applications—whether in metabolic engineering or genetic networks (see <https://www.science.org/doi/10.1126/science.aac7341>). In fact, the OFF states of the shown examples appear to have in most cases a very high basal level, what limit they value as regulatory nodes in logic circuits.
3. Not sure what Authors mean by *domesticating* TFs?

Reviewer #2 (Remarks to the Author):

In this article d'Oelsnitz et al. present a bioinformatics tool for predicting protein DNA interactions for transcriptional regulators. Snowprint seems to be a very nice and potentially useful tool, contributing to a growing collection of similar bioinformatics tools from the Ellington lab. In general, the manuscript is well-written, data of high quality and the tool should be very useful to the broader community. I only have a few suggestions for clarification, and to perhaps better understand the limits of Snowprint.

1. The authors used a variety of loosely defined metrics (e.g., induction range, signal-to-noise ratio, fold range) that make it a bit difficult to appreciate the absolute performances. A note in the supplement (or in the main text) defining the metrics would be most useful.
2. In the benchmark experiments the authors demonstrate that Snowprint predict 67 out of 147 experimentally validated regulator operator pairs. Some readers may find it useful to know why Snowprint failed to predict 80 interactions, and what can be gleaned from these failures in future predictions.
3. The Violin plots in Figure 2b are a bit hard to interpret, for example what is the importance of the

E-value, what does the green shaded area tell us etc. Perhaps additional alternative plots like a series of stacked histograms or kernel density distributions in the supplement would help improve the interpretation of the data. Similar comment for Supplementary Figure 2, though this plot is a bit easier to digest.

4. In Figure 3c, some of the repressed and unrepressed sets are inverted. Why? This seems interesting for many reasons, if the authors could elaborate on the unexpected outcomes and the impact (or limit) of the Snowprint predictions that would be greatly appreciated.

5. In the last section of the results the authors select several compounds (olivetolic acid, geraniol, sitagliptin, ursodiol, tetrahydropapaverine, and artemisinic acid) to induce sets of transcriptional regulators without any preamble. How did the authors know that any of these ligands could potentially work with the selected transcriptional regulators? Are there any control experiments to suggest that induction could not have been achieved by other transcriptional regulators? Is there anything that can be gleaned from mapping a given input to a particular transcriptional regulator? Etc.

Reviewer #3 (Remarks to the Author):

Overall, I appreciated the work presented in this study and I believe it brings a significant advancement to the field, I am therefore in favor of publication. However, I have some suggestions to improve the clarity and depth of the results presented.

1) My main concern related to the results presented in the study, is the lack of comparison with state-of-the-art models that perform the same prediction. I understand that existing models don't necessarily have the same inputs and outputs, but the authors should try to find a way to compare Snowprint's performance with existing models. Indeed, as the study is presented here, it's hard to interpret the performance of Snowprint, especially the 67/147 good predictions, which seem to be a low performance (less than 50%).

2) Authors state that Snowprint's workflow relies on finding the genetic context of a regulator's protein sequence. I think there might be a limit to the use of the model if such context is not available, so the authors should briefly state what the model can do in such cases.

3) The terms "regulator:operator" and "regulator:DNA" are both used to refer to the same pairing, unless I am mistaken. Only one term should be used, or the difference should be clarified. Also, authors sometimes use "regulator-operator", a choice should be made between ":" or "-".

4) The field "synthetic product addiction" is unclear to me, I think this should be briefly described for readers outside the field.

5) The statement "this approach provides little flexibility to tune sensor performance" should either be supported by a reference or further justified.

6) When stating "these models cannot generalize across diverse protein families", the authors should briefly describe why having a model per protein family is really a limit and how the fact of building a protein family-agnostic model is indeed an improvement.

7) "seed operator" should be briefly described when first mentioned.

- 8) The term "remarkable achievement" should be mitigated unless existing models are vastly outperformed by Snowprint in the benchmarking. As there is no comparison presented in the paper here, it's hard to believe this statement.
- 9) The part of the benchmarking results from "The inverted repeat score" to "background signal in genetic circuits" should be clarified, to me it was not obvious to understand and interpret in terms of the model's capabilities.
- 10) The sentence "This ~85% success rate..." is too long, it should be split.
- 11) When writing "twenty-four of the regulator:operator", the authors should clarify from which pairs these 24 have been drawn. I guess it's the correctly predicted pairs, but this should be explicit.
- 12) I believe there is a typo or a missing word in "that an increasing stable of new regulator:operator". Or the sentence should be split in two.
- 13) When mentioning the "dose-response measurements", please state which concentration ranges were tested, why these ranges were selected, and if they are relevant to the stated applications further mentioned in the text.
- 14) I believe there is a typo or a missing word in "signal-to-noise ratios [...] were produced". I would remove 'were produced' and move 'respectively'.
- 15) The sentence "These results were especially surprising..." was unclear to me, the explanations should be slightly extended with another sentence in my opinion.
- 16) On figure 1, the color for inverted repeats should be more striking, I struggled to spot these.
- 17) On figure 2's legend, the authors should specify what scheme corresponds to the regulator and operator. The term "validated-predicted" should be replaced with "predicted" in my opinion.
- 18) The authors should try to find some hypothesis on why some predictions are showing no experimental validation at all (ABB, ACP, KSU, PAT).
- 19) On figure 3, the connectors from panel (a) to other panels were a bit confusing to me, I think the authors should replace those by simply stating next to the compound which regulator is tested in the dose-response curve.
- 20) If possible the authors should extend the dose-response curve for ursodiol to higher concentrations, or state why it has not been done.
- 21) Individual points are not clearly visible on figure 4 panels b-e, I think they should be replaced with mean and standard deviation.
- 22) The last sentence in supplementary figure 5's legend is confusing, it should be rephrased.
- 23) Having full sequences in supplementary tables 3 and 4 does not seem appropriate for reading, I believe some schematics could be more interpretable.

Point-by-point response to Reviewer comments

Statements from the previous version are quoted in blue, and new excerpts are quoted in red.

Key additions to the manuscript include the following:

1. A new Supplementary Figure (Figure S2) that benchmarks our tool against a state-of-the-art model for making similar predictions.
2. A new Supplementary Discussion (Discussion S1) that compares Snowprint to other existing tools that perform similar prediction tasks.
3. A new Supplementary Figure (Figure S9) describing the frequency of failure modes encountered during benchmarking Snowprint, as suggested by Reviewer #2.
4. A new Supplementary Figure (Figure S10) with illustrations of plasmids designs used in this study, as suggested by Reviewer #3.
5. A new Supplementary Note (Note S1) to clarify the definition of terms to describe metrics, as suggested by Reviewer #2.

Reviewer #1

Remarks to the authors:

The article by d'Oelsnitz et al reports a computational method for identifying potential bacterial FT-target promototer pairs that can be leveraged for designing biosensors for small (and not so small) molecules of interest. The background and methodological basis aligns well with similar approaches published in the last few years (see eg <https://doi.org/10.1021/acssynbio.2c00382>, <https://doi.org/10.1021/acssynbio.2c00491>, <https://doi.org/10.1021/acssynbio.2c00679>, <https://doi.org/10.3389/fbioe.2023.1118702>) and therefore the article is one more addition to the arsenal of choices that metabolic engineers and genetic circuit designers have for assembling given constructs.

Overall, the article is well written and the message clear (with some flaws, see below). The platform is experimentally validated and, at least in the examples presented, seems to work to an extent. There are some issues, however, that need to be tackled:

We thank the Reviewer for their appreciation of our work.

Major comments/concerns:

In the general description of approach (summarized in Fig. 1), it is taken for granted that proximity of a TF gene to a metabolic cluster) is an indication that such TF regulates the gene or operon in response to pathway substrates. While this is true in many cases, there are others where this does not happen (see eg a classical discussion on the matter in <https://shorturl.at/bKXY2>). This may explain the somewhat poor predictions (67 out of 147) that could be validated upfront.

This Reviewer brings up an important point regarding possible failure modes for our approach. While we do not make any claims regarding the ability of our tool to predict a transcription factor's "response to a pathway substrate," we agree that a transcription factor's binding sequence is not always located proximal to its coding sequence. As we already suggest in the **Discussion**:

Discussion

One factor that may limit predictive accuracy is the assumption that the target regulator binds to its own promoter, which is true for the majority of prokaryotic regulators, but not all³⁵. (p. 8)

The platform predicts nothing re dynamic range of the transfer input-output functions, an essential parameter for circuit design. In fact the data of Fig. 4 shows very different ranges, which may or not be useful for the envisioned applications—whether in metabolic engineering or genetic networks (see <https://www.science.org/doi/10.1126/science.aac7341>). In fact, the OFF states of the shown examples appear to have in most cases a very high basal level, what limit they value as regulatory nodes in logic circuits.

It is true that the dynamic range for ligand-binding is an important parameter for circuit design and that our tool cannot predict this parameter. However, this was not the goal of our work, and overlooks the most useful features of Snowprint. Our goal was to predict the DNA-binding sequence for bacterial transcription factors in a generalizable way. In consequence, we have in fact found that we were able to produce relatively high dynamic ranges for DNA binding across a panel of extremely diverse TFs without any circuit optimizations (24 of the 33 TFs produced a >20-fold dynamic range for DNA binding, as shown in **Figure 3**). While circuits for other TFs do produce a high basal level in the OFF state, this is not entirely surprising, given that these TFs are sourced from extremely divergent bacterial species, and does not in any way minimize the value of being able to find workable binding sites for further optimization. Snowprint is ultimately a discovery tool. Nonetheless, we agree that the integration of newly discovered biosensors into genetic networks is an important goal and is worthwhile of commentary, which we now include in the discussion:

Discussion

We anticipate that this tool – along with others^{38,42,43} -- will be of great utility for synthetic biologists and bioengineers working to create, improve, or adapt genetic biosensors for applications in high-throughput screening, diagnostics, and genetic circuitry⁴⁴. (p. 9)

*Not sure what Authors mean by *domesticating* TFs?*

We would like to thank this Reviewer for pointing out a potential source of confusion. While we describe our use of the term “domestication” in the **Introduction** ...

*Such applications typically repurpose regulators mined from diverse microbial genomes in a process called “biosensor domestication”, whereby the regulator protein and the regulator-binding sequence, or operator, are retrofitted in a genetic circuit to express a heterologous gene, often in a model organism such as *Escherichia coli*^{5,6}. (p. 3)*

... we also appreciate that this language is not commonly used in the field. Therefore, we have removed the use of this word in sections where it was not also described. Below we detail the edits we have made to address this issue:

Abstract

A panel of the newly ~~domesticated~~ repurposed regulators were then screened for response to biomanufacturing-relevant compounds ... (p. 2)

Reviewer #2

Remarks to the authors:

In this article d'Oelsnitz et al. present a bioinformatics tool for predicting protein DNA interactions for transcriptional regulators. Snowprint seems to be a very nice and potentially useful tool, contributing to a growing collection of similar bioinformatics tools from the Ellington lab. In general, the manuscript is well-written, data of high quality and the tool should be very useful to the broader community. I only have a few suggestions for clarification, and to perhaps better understand the limits of Snowprint.

We thank the Reviewer for their appreciation of our work.

Major comments/concerns:

1. The authors used a variety of loosely defined metrics (e.g., induction range, signal-to-noise ratio, fold range) that make it a bit difficult to appreciate the absolute performances. A note in the supplement (or in the main text) defining the metrics would be most useful.

We appreciate the potential source of confusion this Reviewer brings to our attention. We have made two systematic changes to the manuscript to address this issue. First, we standardize the language used to describe metrics. Throughout the paper, the term “signal-to-noise ratio” is revised to “dynamic range”, while “induction range” and “fold range” are revised to “induction ratio” These terms are now defined in a new **Supplementary Note (Supp Note 1)**.

Supplementary Note 1: Definitions

Dynamic range

We define the dynamic range as the fluorescent signal produced by the un-repressed promoter (without expression of the cognate regulator), divided by the fluorescent signal produced by the same promoter when repressed (with expression of the cognate regulator).

Induction ratio

We define the “induction ratio” as the highest fluorescent signal of the cell population upon induction with the target ligand divided by the basal fluorescence of the cell population without induction with any ligand.

(p. 16)

Manuscript changes to standardize the use of these defined terms are listed below.

Introduction

... of which 28 were able to effectively repress GFP expression within *E. coli* and 24 achieved over a 20-fold ~~dynamic range~~ ~~signal-to-noise ratio~~. The top 24 newly identified regulatory pairs were screened for response to industrially relevant biosynthetic intermediates, and novel genetic

biosensors for olivetolic acid, geraniol, ursodiol, and tetrahydropapaverine were found, achieving induction ratios of 6-, 3.6-, 2.3-, and 10.7-fold, respectively (see **Supplementary Note 1** for definitions). (p. 4)

Abstract

... of which 28 were shown to influence gene expression and 24 produced a >20-fold dynamic range signal-to-noise ratio. ... and alkaloid (tetrahydropapaverine) with induction ratios up to 10.7-fold. (p. 2)

Results

Twenty four of the regulator:operator pairs with the highest dynamic ranges signal-to-noise ratios (>20-fold, average of 64-fold) ... (p. 7)

Sigmoidal transfer functions characteristic of transcriptional regulators were observed (**Figure 4b**), and induction ratios signal-to-noise ratios reached 10.7-, 6.0-, 3.6-, and 2.3-fold for tetrahydropapaverine, olivetolic acid, geraniol, and ursodiol, respectively, were produced. (p. 7)

Discussion

Among the 33 predictions used to create GFP reporter circuits in *E. coli*, 28 were able to modulate gene expression by over 1.5-fold, among which the top 24 circuits produced a dynamic range signal-to-noise ratio over 20-fold. (p. 8)

Figure 4a legend:

Data represents the induction ratio fold-change in fluorescence over *E. coli* cells bearing the identical plasmids induced with DMSO only, performed in biological triplicate (p. 14)

Methods (“Biosensor response assay”)

This assay format was used to determine regulator dynamic range signal-to-noise ratios (**Figure 3**), as well as regulator screening and dose-response measurements (**Figure 4**). (p. 21)

2. In the benchmark experiments the authors demonstrate that Snowprint predict 67 out of 147 experimentally validated regulator operator pairs. Some readers may find it useful to know why Snowprint failed to predict 80 interactions, and what can be gleaned from these failures in future predictions.

We agree with this Reviewer that readers would benefit from an analysis of failure modes that were encountered when benchmarking Snowprint. We have therefore created a new Supplementary Figure that describes three failure modes: (1) the operator is not found in the extracted inter-operon region, (2) the operator is within the extracted inter-operon region but the search algorithm failed to find it, and (3) too few homologs were identified and thus a confident operator prediction could not be generated. This entire analysis uses data presented in **Supplementary Data 1**, which contains all Snowprint benchmarking inputs and metrics. In addition, we have referenced our new Supplementary Figure in the **Discussion** section.

Discussion

This analysis revealed a coarse correlation (**Supplementary Figure 8**) suggesting that if the known operator is found in the upstream, inter-operon region, then the chances of it being correctly predicted increase. Upon further analysis, the most frequent failure mode was caused by the operator sequence not being found within the extracted inter-operon region (48/80), followed by the search algorithm not identifying the appropriate operator sequence (22/80). The least frequent failure mode was caused by low quality predictions resulting from too few homologs being available (10/80) (**Supplementary Figure 9**). (p. 8)

Supplementary Figure 9. Frequencies of failure modes encountered during benchmarking

Snowprint failure modes were binned into three categories: (1) the operator is not found within the inter-operon region, (2) the search algorithm failed to identify the validated operator, and (3) the prediction quality is low due to fewer than 10 homologs being used to create the motif. To be binned into category (1), the validated operator must not align to the extracted inter-operon region (alignment score lower than 90/100). To be binned into category (2), the validated operator must align nearly perfectly to the inter-operon region (alignment score of >90/100), but align poorly to the predicted operator (alignment score of <60/100). All benchmarking metrics used to create this plot can be found in **Supplementary Data 1**.

(p. 10)

3. The Violin plots in Figure 2b are a bit hard to interpret, for example what is the importance of the E-value, what does the green shaded area tell us etc. Perhaps additional alternative plots like a series of stacked histograms or kernel density distributions in the supplement would help improve the interpretation of the data. Similar comment for Supplementary Figure 2, though this plot is a bit easier to digest.

We thank this Reviewer for highlighting an area of potential confusion. **Figure 2b** can best be interpreted in terms of **Supplementary Table 1**, which shows three categories for the alignments of predicted operators to their validated operator counterparts: (1) very low E-values, (2) very high E-values, and (3) E-values close to the 0.01 cutoff for the green shaded area in **Figure 2b**. This Table illustrates that predicted operators with an E-value less than 0.01 closely resemble the validated operator and are therefore likely physiologically relevant.

Protein ID	E-value	Alignment		
WP_001224188.1	3.7*10 ⁻¹³	Documented	1 gttttataATAAACGGAgagttaTCCGTTTGTcaa	35
		Predicted	1 -----ataatAAACGGATAGTTATCCGTTTgtcaa	30

AEM66515.1	0.0012	Documented	1 TAGCCACGTCTGGACAAAGTGAGAGATCGTGTCTAGACAACGCCACGGTT	50
		Predicted	1 -----gATcGtGtCTAGACAacgcc-----	20
NP_415533.1	0.0076	Documented	1 TGTA A A A T T T G A C C A T T T G G T C C A C T T T T T T C T	33
		Predicted	1 -T T A A C T T T T A A A A C T G G C -----	20
WP_011060270.1	0.0064	Documented	1 -----T C A A A C A A G T G T T T G T C A G G ---	20
		Predicted	1 aagccTGAACGTATGTT--TCAaaca	26
WP_005058758.1	0.0072	Documented	1 -----C T T A A C G C A T G G C A T G T G C G T T A T G	25
		Predicted	1 ggtagCGTGC GTTACGCACGtcata-----	25
CAY46636.1	0.0049	Documented	1 ---GCT-----TGTATGTACAAGT--	16
		Predicted	1 acgtcGTATATGTATATACAaataa	26
NP_391277.1	0.0061	Documented	1 ---A A A T T T G T C C G T A T A C A T T T T --	21
		Predicted	1 atttaATTAGTACGTACAATataga	26
NP_227848.1	0.0018	Documented	1 --A A T T T C T T T C T G A G -G A A G A T A G A	23
		Predicted	1 agacaTTCTCAAAGTGAGAAgaatg-	25
WP_003084488.1	1.0	Documented	1 tttcTAG--AGTAaaTACTCTAaagt--	24
		Predicted	1 --tcacgCGGGGCGCGCCCGCactgt	26

Supplementary Table 1. Global alignment of documented and predicted operators.

The first and last alignments display excellent and poor scores for comparison, respectively. All other alignments have E-values between 0.01 and 0.001. The pairwise EMBOSS needle algorithm was used to create alignments with default parameters. Documented operators represent sequences experimentally validated to bind the corresponding protein regulator. Refer to Supplementary Data 1 for more information on sequence metrics and sources.

In the manuscript, we reference this **Supplementary Table** to better explain why we chose an E-value of < 0.01 as likely physiologically relevant, shortly before more thoroughly describing **Figure 2b**:

Results: “Benchmarking Snowprint performance”

Operator predictions were considered to be significant if the E-value produced by aligning the predicted sequence to the known operator was less than 0.01 (see **Supplementary Table 1**)

We found that Snowprint was able to identify operators that were significantly similar to known, experimentally validated operators (E-value < 0.01) for 58%, 50%, 44%, 42%, and 35% of regulators across the TetR, IclR, MarR, LacI, and GntR structural families, respectively (**Figure 2b**). (p. 5)

Figure 2b

b
We now reference **Supplementary Table 1** in the legend of **Figure 2**:

Figure 2 legend:

(b) Similarity scores for predicted operators among several structural regulator families. The E-value of 0.01 was used as a threshold to indicate significance (**Supplementary Table 1**). (p. 12)

We believe that producing a stacked histogram of **Figure 2b** as an additional supplementary figure would be redundant, since a stacked histogram plot is essentially equivalent to our violin plots (ala the green shading now shown in **Figure 2b**). We would also like to note that all data used to generate the plots in **Figure 2b** and **Supplementary Table 1** are included in **Supplementary Data 1**.

4. In **Figure 3c**, some of the repressed and unrepressed sets are inverted. Why? This seems interesting for many reasons, if the authors could elaborate on the unexpected outcomes and the impact (or limit) of the Snowprint predictions that would be greatly appreciated.

We agree that the observation of one regulator's response being inverted is interesting and warrants further discussion. While uncommon, some TetR-family regulators have been shown to act as activators rather than repressors by directly interacting with the RNA polymerase core enzyme (de Souza Pinto Lemgruber, R., Valgepea, K., et al., 2019). This mechanism, however, is unlikely to explain our observation of an inverted response for the RBP44292.1 regulator, since it originates from the organism *Roseimicrobium gellanilyticum*, which belongs to a different phylum than *Escherichia coli*, and thus is less likely to interact with the *E. coli* polymerase. Alternatively, RBP44292.1 may induce a physical change in DNA topology that promotes transcription, as has been seen in other regulator families (E. E. Z. Heldwein & R. G. Brennan, 2001). We have now included a discussion on these issues in the manuscript.

Discussion

Among the five regulators that showed no ability to repress transcription, it is possible that the Snowprint-predicted operator is not bound by the regulator, or alternatively, the regulator might not express or fold appropriately in *E. coli*. Interestingly, one regulator (RBP44292.1) produced an inverted response, similar to activator proteins. This may be due to making direct contacts with RNA polymerase or via altering DNA topology to promote transcription, which has been observed in related transcription factor families^{36,37}. (p. 8)

5. In the last section of the results the authors select several compounds (olivetolic acid, geraniol, sitagliptin, ursodiol, tetrahydropapaverine, and artemisinic acid) to induce sets of transcriptional regulators without any preamble. How did the authors know that any of these ligands could potentially work with the selected transcriptional regulators? Are there any control experiments to suggest that induction could not have been achieved by other transcriptional regulators?

We agree that the rationale for choosing the target ligands for screening could be described in more detail. Accordingly, we have added text to the **Results**.

Results: “Discovering novel biosensors for biomanufacturing-relevant ligands”

The compounds olivetolic acid, geraniol, sitagliptin, ursodiol, tetrahydropapaverine, and artemisinic acid at 100 uM concentration were used as test ligands, as each one is a pharmaceutical, or intermediate thereof, that has been produced from an engineered microbe or enzyme^{25,30–34}. In addition, we chose these compounds due to their lipophilicity, since highly promiscuous regulators tend to have large hydrophobic binding cavities³⁶. (p. 7)

Regarding control experiments, in **Figure 4a** we show that no significant change in GFP expression results when cells are induced with 100 uM of the target ligand, unless the cells already contain a responsive regulator, which occurred in 12 of the 120 tested conditions.

“Is there anything that can be gleaned from mapping a given input to a particular transcriptional regulator? Etc.”

Given that the regulators screened in **Figure 4** are largely uncharacterized and are thought to bind to structurally diverse ligands (based at least on their genomic context), we hesitated to speculate on mechanisms of ligand recognition or any ligand-protein “mapping” without collecting additional experimental data, which we believe would be outside the scope of this study. However, we agree that it would be extremely interesting to explore such a mapping in follow-up studies.

Reviewer #3

Remarks to authors:

Overall, I appreciated the work presented in this study and I believe it brings a significant advancement to the field, I am therefore in favor of publication. However, I have some suggestions to improve the clarity and depth of the results presented.

We thank the Reviewer for their appreciation of our work.

Major comments/concerns:

My main concern related to the results presented in the study, is the lack of comparison with state-of-the-art models that perform the same prediction. I understand that existing models don’t necessarily have the same inputs and outputs, but the authors should try to find a way to compare Snowprint’s performance with existing models. Indeed, as the study is presented here, it’s hard to interpret the performance of Snowprint, especially the 67/147 good predictions, which seem to be a low performance (less than 50%).

While performing a direct comparison of our tool with state-of-the-art models is difficult due to the incompatibility of model input types (as this Reviewer also acknowledges), we have nonetheless made these comparisons to the best of our ability. In particular, we compare Snowprint to a recently described statistical model for predicting TetR binding sequences (Long et al., 2020). Operator predictions generated by Snowprint and the statistical model were aligned to the experimentally validated operator sequences. This alignment is now a new **Supplementary Figure**, and demonstrates that the predictive accuracy of both models are comparable for TetR-family proteins; 29/50 Snowprint predictions are significantly similar to the validated operator (E-value < 0.01), compared to 31/50 for the statistical model. For some TetR proteins, the more general Snowprint analysis outperformed the statistical model, even though the latter was specifically trained on TetR-family members (and of course, the statistical model failed to return a prediction for all proteins that did not belong to the TetR family). These new results further highlight the utility of Snowprint as a protein-agnostic tool.

Supplementary Figure 2. Comparing the ability of Snowprint and a published statistical model to predict TetR operators. Predictions were generated for 50 TetR-family proteins using Snowprint and a statistical model published by Long et al. (a) The E-value of the predicted operator aligned to the experimentally validated operator is plotted for each TetR-family protein. See **Supplementary Data 2** for all model inputs, predicted operators, and E-values used to generate this plot. (b) Violin plots describing the global distribution of validated:predicted operator E-values between the two models. For both (a) and (b) blue represents Snowprint predictions and black represents statistical model predictions.

(p. 3)

In addition, we include a new **Supplementary Discussion 1**, where we provide a rationale for why Snowprint could not be readily compared to many other published models that generate predictions for transcription factor binding sites.

Supplementary Discussion 1: Comparison to other prediction tools

While several other existing tools also predict a transcription factor's DNA binding sequence, these tools rarely use the same inputs as Snowprint, making direct comparison virtually impossible. Below we comment on several other existing tools that make similar predictions

DMINDA 2.0

Incomparable input requirements: a set of promoter sequences in FASTA format

DeepGRN

Incomparable input requirements: CHIP-Seq and RNA-Seq datasets

(p. 16-17)

Finally, we have added to the **Results** section, referencing the new figures, dataset, and discussion.

Results: "Benchmarking Snowprint performance"

We found that Snowprint was able to identify operators that are significantly similar to known, experimentally validated operators (E-value < 0.01) for 58%, 50%, 44%, 42%, and 35% of regulators across the TetR, IclR, MarR, LacI, and GntR structural families, respectively (Figure 2b). Although many existing tools for predicting transcription factor binding sites could not be compared to Snowprint due to substantial differences in the format of the input data (see **Supplementary Discussion 1**), we were able to compare Snowprint to a recently described statistical model train to predict TetR binding sites²¹. Our analysis indicates that the predictive accuracy of Snowprint was similar to the statistical model, with 31/50 operators producing an E-value of less than 0.01, compared to 29/50 for Snowprint (**Supplementary Figure 2, Supplementary Data 2**). However, in some cases Snowprint outperformed the statistical model, even though the latter was specifically trained on TetR-family members (**Supplementary Figure 2a**). Furthermore, the statistical model failed to return a prediction for proteins that did not belong to the TetR family, which was not the case for Snowprint (**Figure 2b**). (p. 5)

2) Authors state that Snowprint's workflow relies on finding the genetic context of a regulator's protein sequence. I think there might be a limit to the use of the model if such context is not available, so the authors should briefly state what the model can do in such cases.

While we have found this failure mode to be quite rare, it is nonetheless possible that genetic context may not be available for a given target regulator. In this case, the user may input a highly similar regulator sequence, which will likely produce a nearly identical operator prediction since the collection of homologs used to generate the predicted operator motif will be much the same. We have added text to the **Discussion** to mention this possible failure mode and suggest a course of action if this issue is encountered by the user. We would also like to thank this Reviewer for bringing this point to our attention, and we strongly agree that it deserves mentioning in the manuscript.

Discussion

This analysis revealed a coarse correlation (Supplementary Figure 8) suggesting that if the known operator is found in the upstream, inter-operon region, then the chances of it being correctly predicted increase. ... While not encountered during benchmarking, another possible failure mode is that the regulator's

genetic context might not be available. In this case a highly homologous regulator may be used as an input, which will likely produce a nearly identical operator prediction as the target regulator, since the collection of homologs used to generate the predicted operator motif will be much the same. (p. 8)

3) The terms “regulator:operator” and “regulator:DNA” are both used to refer to the same pairing, unless I am mistaken. Only one term should be used, or the difference should be clarified. Also, authors sometimes use “regulator-operator”, a choice should be made between “:” or “-”.

We thank this keen observation made by this Reviewer. All instances of “regulator:DNA” and “regulator-operator” have been changed to “regulator:operator” throughout the manuscript.

Abstract

Here we present Snowprint, a simple yet powerful bioinformatic tool for predicting regulator:operatorDNA interactions. (p. 2)

Results: “Benchmarking Snowprint performance”

To benchmark the performance of Snowprint, we first curated a list of 147 experimentally validated regulator:operator pairs from the literature (p. 5)

Discussion

... genome and protein sequences to predict transcriptional regulator:operatorDNA interactions. (p. 7)

4) The field “synthetic product addiction” is unclear to me, I think this should be briefly described for readers outside the field.

We agree that this term is not common in the field and have therefore removed it from the **Introduction**. We believe that the “synthetic product addition” sub-field fits within the field of “dynamic regulation” that is already mentioned in the manuscript. We thank this reviewer for their clarifying suggestion.

Introduction

Ligand-inducible transcriptional regulators are becoming indispensable biosensors for synthetic biology and bioengineering applications, such as high-throughput screening, dynamic regulatory circuits, and diagnostics ~~synthetic product addiction~~¹⁻⁴. (p. 3)

5) The statement “this approach provides little flexibility to tune sensor performance” should either be supported by a reference or further justified.

This Reviewer brings up a valid argument that further justification is needed to back one of our claims. We have since added text to further justify our claim and reference a paper that creates a synthetic ligand-responsive promoter in which previous attempts to use the native promoter sequence failed.

Introduction

In consequence, this approach provides little flexibility to tune sensor performance, since cryptic promoters or inhibitory noncoding regions may produce high basal signals or dampen a sensor’s responsiveness⁷. (p. 3)

6) When stating “these models cannot generalize across diverse protein families”, the authors should briefly describe why having a model per protein family is really a limit and how the fact of building a protein family-agnostic model is indeed an improvement.

We thank this Reviewer for their excellent suggestion. We have since added text to highlight the importance of building a protein-agnostic model for predicting transcription factor-DNA interactions:

Abstract

Snowprint represents a unique, protein-agnostic generalisable tool that greatly facilitates the discovery of ligand-inducible transcriptional regulators for bioengineering applications. (p. 2)

Introduction

Alternatively, statistical learning and structural information has also been used to create models capable of predicting transcription factor binding sites, but these models cannot generalize across diverse protein families¹⁶. As over 71 transcription factor structural families have been reported to date¹⁷, a generalizable tool for predicting binding sequences would enable biotechnologists to identify and repurpose many more transcription factors. (p. 3)

Discussion

Snowprint is a protein-agnostic simple yet powerful tool that leverages rapidly growing public databases of genome and protein sequences to predict transcriptional regulator:DNA interactions. (p. 7)

7) “seed operator” should be briefly described when first mentioned.

While we believe that we have appropriately defined the “seed operator” in the text, we agree that the language used could be edited to improve clarity. Therefore, we have made the following edit:

Results: “Workflow for operator prediction”

The sequence with the highest inverted repeat score, which we term a “seed operator”, is then used for the subsequent step. (p. 4)

8) The term “remarkable achievement” should be mitigated unless existing models are vastly outperformed by Snowprint in the benchmarking. As there is no comparison presented in the paper here, it’s hard to believe this statement.

We thank this Reviewer for their comment and have removed this phrase from the manuscript.

Results: “Benchmarking Snowprint performance”

We found that Snowprint was able to identify operators that were significantly similar to known, experimentally validated operators (E-value < 0.01) for 58%, 50%, 44%, 42%, and 35% of regulators across the TetR, IclR, MarR, LacI, and GntR structural families, respectively (Figure 2b), a remarkable achievement that bodes well for the assured identification of new operators. (p. 5)

9) The part of the benchmarking results from “The inverted repeat score” to “background signal in genetic circuits” should be clarified, to me it was not obvious to understand and interpret in terms of the model’s capabilities.

We agree with this Reviewer that some sentences describing model predictions should be edited for clarity. We have therefore made the following edit to the manuscript:

Results: “Benchmarking Snowprint performance”

The inverted repeat score (see **Methods**) and the number of homologs used to generate the predicted operator were also collected and indicate a quality level for each prediction, since the confidence in a prediction is correlated with the number of homologs assessed and the length of the inverted repeat ~~can be found in~~ (**Supplementary Data 1**). (p. 5)

Nonetheless, we believe that the other sentences within this paragraph do adequately describe the model’s performance. The sentences that follow indicate that Snowprint can predict experimentally validated operators across several protein structural families in 35-58% of cases, depending on the family. Also, the generalizability of Snowprint to diverse phylogenetic spaces is clearly described. Finally, we suggest that Snowprint is able to identify more symmetric operators compared to their natural counterparts, which may enable stronger repression in genetic circuits. We believe that all three of these points are succinct and clear in describing the model’s capabilities.

10) The sentence “This ~85% success rate...” is too long, it should be split.

We agree that splitting this sentence into two will help to improve the readability of our manuscript, and we thank this Reviewer for suggesting this change. We have edited the text accordingly, as follows.

Results: “Using Snowprint to domesticate generalist transcription factors”

This ~85% success rate is quite surprising, given that the regulators were sourced from wildly divergent microbial hosts and that synthetic promoters were crafted in accord with simple rules. ~~and~~ Furthermore, *in vivo* repression could fail for reasons other than correct regulator:operator matching, such as issues relating to heterologous protein production. (p. 6)

11) When writing “twenty-four of the regulator:operator”, the authors should clarify from which pairs these 24 have been drawn. I guess it’s the correctly predicted pairs, but this should be explicit.

We appreciate the excellent suggestion made by this Reviewer, and we agree that the 24 pairs used for ligand screening could be indicated more explicitly. To address this issue, we have highlighted these 24 regulators in **Supplementary Table 2** and have edited the **Results** section to reference this Table.

Results: “Discovering novel biosensors for biomanufacturing-relevant ligands”

Twenty four of the regulator:operator pairs with the highest dynamic ranges (>20-fold, average of 64-fold) were screened for de-repression (see **Supplementary Table 2**). (p. 7)

Supplementary Table 2 (truncated)

Genbank ID	Number of aligned sequences	Consensus Score	Predicted operator
ANI80042.1	7	94.99	gttcgAAAATAAACAATCTTGTTTCTTTTattca
AOO80349	12	89.099	cctgcTAAACTAAACGGTTCAGTTTAttcgc
AOR61801.1	10	62.821	tatacCCATAATTAATGACCCAGAAGTCATTAATTATGGtgtcc
EEI22781.1	38	95.314	ctcatAGTGACACTGTGTCACTcgatt
ABB11310.1	24	85.375	cgattTGC GTGCCCGGGTGCCGGGGCACGCAtcctg
ACP22327	15	75.794	cttatTAGATTTCAATTGACATCTAtatat
AEI82798.1	38	71.203	cttTGCGGTCCcGGACCGCAtga
ASG24684.1	9	100	ggactGTA CTGTATGATACAGTACgtaat
AMY71450	11	82.012	caatcACTGTACCGTCCAGACGGTAAAGTcaacc

Supplementary Table 2 legend:

... Results and metadata associated with Snowprint predictions of TetR-family transcriptional regulators targeted for testing within *E. coli*. Regulators that produced a >20-fold dynamic range in *E. coli* are highlighted green. ... (p. 13)

12) I believe there is a typo or a missing word in “that an increasing stable of new regulator:operator”. Or the sentence should be split in two.

We agree that the wording of this long sentence could potentially confuse readers. We appreciate the suggestion made by this Reviewer and have split the sentence into two to improve readability:

While the initial responses were in the 1.25- to 3-fold range, the fact that any of these uncharacterized regulators displayed a response was both surprising, and validated the hypothesis. These results highlight that an increasing stable of new regulator:operator pairs supports the provides a valid starting point for identification of biosensors for a diversity of chemical effectors. (p. 7)

13) When mentioning the “dose-response measurements”, please state which concentration ranges were tested, why these ranges were selected, and if they are relevant to the stated applications further mentioned in the text.

We thank the Reviewer for their comment. We have added a sentence to the Results section to indicate the concentration ranges used for the dose-response measurements, as well as a rationale for choosing these ranges.

To further validate these newly identified interactions, dose-response measurements were carried out with those pairs that produced the highest signals with geraniol, olivetolic acid, ursodiol, and tetrahydropapaverine. Tested ligand concentrations ranged from 10 uM to 5 mM, depending on the ligand’s solubility limit in 1% DMSO. (p. 7)

14) I believe there is a typo or a missing word in “signal-to-noise ratios [...] were produced”. I would remove ‘were produced’ and move ‘respectively’.

We greatly appreciate the keen observation made by the Reviewer and have corrected this sentence to remove the typo.

Results: “Discovering novel biosensors for biomanufacturing-relevant ligands”

Sigmoidal transfer functions characteristic of transcriptional regulators were observed (Figure 4b), and induction ratios reached 10.7-, 6.0-, 3.6-, and 2.3-fold for tetrahydropapaverine, olivetolic acid, geraniol, and ursodiol, respectively, ~~were produced~~. (p. 7)

15) *The sentence “These results were especially surprising...” was unclear to me, the explanations should be slightly extended with another sentence in my opinion.*

We agree that this sentence could be clarified. We now appreciate the possible source of confusion and have in turn edited the sentence:

These results were especially surprising, given that the prediction quality metrics, such as number of homologs used and conservation scores, were generally worse for the domesticated sensors relative to the validated regulator:operator pairs in the benchmarking dataset (Supplementary Table 2, Supplementary Data 1). (p. 8)

16) *On figure 1, the color for inverted repeats should be more striking, I struggled to spot these.*

To improve clarity, we have changed the color for the inverted repeats from blue to pink.

17) On figure 2's legend, the authors should specify what scheme corresponds to the regulator and operator. The term "validated-predicted" should be replaced with "predicted" in my opinion.

The legend of **Figure 2** has been updated to clarify the scheme for the operator and regulator, and the term "validated-predicted" was rewritten as "predicted", as suggested by this Reviewer.

(a) **Benchmarking workflow.** Experimentally validated operator regulator pairs are collected from the literature, and regulators for each pair are passed through the Snowprint workflow. Predicted operators are then compared to validated operators. **The regulator is colored purple and the operator is colored blue** (b) Similarity scores for ~~validated-predicted~~ operators among several structural regulator families. The E-value of 0.01 was used as a threshold to indicate significance. The "Other" group contains regulators from the MerR, ArsR, PadR, TrpR, and ROK structural families. **(p. 12)**

18) The authors should try to find some hypothesis on why some predictions are showing no experimental validation at all (ABB, ACP, KSU, PAT).

We agree with this Reviewer that it would be worthwhile to speculate on possible reasons why some transcription factors are apparently not able to repress transcription from their designed promoters (**Figure 3**). Accordingly, we have to the **Discussion** section:

Discussion

Among the 33 predictions used to create GFP reporter circuits in *E. coli*, 28 were able to modulate gene expression by over 1.5-fold, among which the top 24 circuits produced a dynamic range over 20-fold. Among the five regulators that showed no ability to repress transcription, it is possible that the Snowprint-predicted operator was not bound by the regulator, or alternatively, the regulator might not have expressed or folded appropriately in *E. coli*. (p. 8)

19) On figure 3, the connectors from panel (a) to other panels were a bit confusing to me, I think the authors should replace those by simply stating next to the compound which regulator is tested in the dose-response curve.

We appreciate the suggestion made by this Reviewer, which we have now incorporated into a revised **Figure 4**.

20) *If possible the authors should extend the dose-response curve for ursodiol to higher concentrations, or state why it has not been done.*

We could not test higher concentrations of Ursodiol since it is poorly soluble in aqueous solutions. To clarify this point, we have made the following changes to the legend of **Figure 4**.

Figure 4 legend

Dose response measurements for the BAK71752.1, SMC09139, SEE04737, and SMC09139 regulators with tetrahydropapaverine (THP), geraniol, olivetolic acid, and ursodiol, respectively. The maximum ligand concentration was chosen based on the compound's solubility limit in 1% DMSO. Assays were performed in biological triplicate and individual data points are shown. (p. 14)

21) *Individual points are not clearly visible on figure 4 panels b-e, I think they should be replaced with mean and standard deviation.*

While journal policies require that individual data points are shown in Figures, we recognize that these data points may be difficult to discern. Therefore, we have increased the height of each subfigure from 12x8 to 10x10 to better display the data points. We also recognize that there was an issue with our plotting software that caused only two of the three data points to be visible in these subplots, and we have now corrected that issue. Please refer to **Reviewer #3**, comment # 19 for the updated **Figure 4**.

22) *The last sentence in supplementary figure 5's legend is confusing, it should be rephrased.*

The last sentence in the legend for **Supplementary Figure 5** (now **Supplementary Figure 6**) is "Error bars represent the standard deviation +/- the mean", which we had thought was a common description. However, we understand that the phrasing of this figure legend could be improved. Thus, we have made the following changes to improve clarity for the reader.

Supplementary Figure 6 legend

Fluorescence of *E. coli* cells containing a reporter plasmid expressing GFP from a promoter bearing a Snowprint-predicted operator (see Figure 3, Supplementary Figure 4) and a regulator plasmid expressing a transcription factor (see Figure 3, Supplementary Figure 10) from a constitutive plasmid. The GFP-expressing promoter contains a Snowprint-predicted operator (see Supplementary Figure 5). In (a) a regulator plasmid expresses the transcription factor predicted to bind to the Snowprint-predicted operator. In (b) the regulator plasmid expresses the CamR transcription factor, which serves as a control transcription factor that should not bind to the promoter driving GFP expression. Assays were performed in biological triplicate. Individual data points are shown in pink. Error bars represent the standard deviation +/- the mean. (p. 7)

23) *Having full sequences in supplementary tables 3 and 4 does not seem appropriate for reading, I believe some schematics could be more interpretable.*

We agree that having schematics of the plasmid designs could be more interpretable for the reader. We have since added a new **Supplementary Figure 10**, which includes a schematic, and an associated legend that refers the reader to other supplementary items that include the promoter architecture (**Figure S4**), operator sequences (**Table S2**), and full plasmid sequences (**Table S3** and **S4**).

Supplementary Figure 10. Schematics of the two circuits architectures used in this study

The pink regulator gene and regulated promoter sequences change for each regulator being tested. Please refer to **Supplementary Figure 5** for an in-depth illustration of the Regulated Promoter region. Please refer to, **Supplementary Table 2** for the GenBank IDs and predicted operator sequences of all regulators. Please refer to **Supplementary Table 3 & 4** for the full plasmid sequence of the Reporter vector and Regulator vector, respectively. (p. 11)

Other updates

To improve the accessibility of Snowprint, we are excited to report that we have developed a user-friendly web application, available at snowprint.groov.bio. The web interface uses a RefSeq ID, a Uniprot ID, or a protein sequence as an input, and returns the same output format produced by our command line tool. This web application uses the DIAMOND software instead of BLAST, and thereby allows for much faster response -- typically 5 minutes instead of 15. To incorporate this new web application into this manuscript, we made the following text additions:

Abstract

Snowprint represents a unique, protein-agnostic tool that greatly facilitates the discovery of ligand-inducible transcriptional regulators for bioengineering applications. A web-accessible version of Snowprint is available at snowprint.groov.bio. (p. 2)

Results: “Workflow for operator prediction”

The consensus operator and conservation scores are displayed in a browser using an interactive React webpage (Supplementary Figure 1), or via a web-accessible version available at snowprint.groov.bio. (p. 4)

Methods: “Building the Snowprint web application”

The web application was split into frontend design and backend data architecture. The frontend was written in Javascript using the React and Material UI libraries, and is hosted using AWS Amplify. The backend ported all code from the command line tool used for benchmarking (<https://github.com/simonsnitz/Snowprint>) into a docker container hosted on AWS Fargate. The only difference between the command line tool and the web application is that the former uses NCBI’s BLAST

to collect protein homologs while the latter uses DIAMOND, which performs faster. While results returned from the command line tool and web application are similar, they may not be identical. (p. 18)

Code availability

Snowprint is Open Access under an MIT License. Source code for the frontend and backend of the Snowprint web application (snowprint.groov.bio) are maintained in the GitHub repositories located at <https://github.com/simonsnitz/snowprint-ui> and <https://github.com/simonsnitz/snowprint-backend> , respectively. (p. 18)

REVIEWERS' COMMENTS:

Reviewer #1 (Remarks to the Author):

Authors have met my concerns in the revised ms.

Reviewer #2 (Remarks to the Author):

All of my concerns have been addressed, I do not have any additional requests.

I highly recommend publication.

Reviewer #3 (Remarks to the Author):

These authors have done an excellent job revising their manuscript and have thoroughly answer all my questions and concerns, adding new materials in the main text and Supplementary Information. I thank them for that and have no further questions to raise.